

# Response of near-inertial energy to a supercritical tropical cyclone and jet stream in the South China Sea: modeling study

Hiu Suet Kung, Jianping Gan*

Department of Ocean Science and Department of Mathematics, Hong Kong University of Science and Technology, Hong Kong

Corresponding author: magan@ust.hk





**ABSTRACT**
We used a well-validated three-dimensional ocean model to investigate the process of energetic
response of near-inertial oscillations (NIOs) to a tropical cyclone (TC) and strong background jet
stream in the South China Sea (SCS). We found that the NIO and near-inertial kinetic energy
(KEni) varied distinctly during different stages of the TC forcing, and the horizontal and vertical
transport of KEni was largely modulated by the velocity and vorticity of the jet stream. The KEni
reached its peak value within ~one-half the inertial period after the initial TC forcing stage in the
upper layer, decayed quickly by one-half in the next two days, and further decreased in a slower
rate during the relaxation stage of the TC forcing. Analyses of the KEni balance indicate that the
weakened KEni in the upper layer during the forcing stage was mainly attributed to the downward
KEni transport due to pressure work through the vertical displacement of isopycnal surfaces, while
upward KEni advection from depths also contributed to the weakening in the TC-induced
upwelling region. In contrast, during the relaxation stage as TC moved away, the effect of vertical
advection on KEni reduction was negligible and the KEni was chiefly removed by the outward
propagation of inertial-gravity waves, horizontal advection and viscous dissipation. Both the
outward wave propagation and horizontal advection by the jet stream provided the KEni source in
the far-field. During both stages, the negative geostrophic vorticity south of the jet stream
facilitated the vertical propagation of inertial-gravity waves.



## 1 Introduction

Near-inertial oscillations (NIOs), whose frequencies are close to the local inertial frequency contain around half of the observed internal wave kinetic energy in the ocean (Simmons and Alford, 2012). NIOs also greatly affect the kinetic energy budget in the deeper ocean as they propagate downward from the surface and enhance the mixing by increasing vertical shear (Gill, 1984; Gregg et al., 1986; Ferrari and Wunsch, 2009; Alford et al., 2016).

Tropical cyclones (TCs), with the rapid change of wind stress, provide an important generation mechanism for the NIOs. Observational studies related to a single storm or tropical cyclone (Price, 1981; Shay and Elsberry, 1987; D'Asaro et al., 1995) showed that the NIOs related to TCs can be a factor 2-3 larger than the background NIOs and last for more than 5 inertial periods (IPs). Using a hurricane-ocean coupled model, Liu et al. (2008) estimated that the energy input of tropical cyclones into the near-inertial currents was about 0.03 TW, about 10% of the total wind-induced near-inertial energy (Watanabe and Hibiya, 2002; Alford, 2003). The input of wind energy to the near-inertial band is also controlled by the translation speed of the TC ($U_h$) (Geisler, 1970; Price, 1981). In fact, the NIOs are largely variable during the forcing and relaxation stage of the TC forcing related to the intensity and translation speed of the TC. The variation is determined not only by the different magnitude of input of wind energy, but also by the different dynamic conditions that regulate the near-inertial kinetic energy (KEni) transport during these stages. The variable response of NIOs during different stage of TC forcing is critical for understanding the process and physics of NIOs.

Once generated, the characteristics of NIOs, in terms of the decay time scale, propagation direction, and propagation speed are influenced by various mechanisms. In linear wave theory, the β-effect leads to an equatorward propagation of the NIOs and reduces its decay time scale because





of the vertical propagation into deeper ocean (Gill, 1984; D'Asaro, 1989; Garrett, 2001).
Background flow fields also impose large influence on the evolution of NIOs. The influence of
background vorticity on NIOs has been observed by Weller (1982) and Kunze and Sanford (1984),
and analytical proofs by Kunze (1985), Young and Ben Jelloul (1997), and Danioux et al. (2015).
Numerical study using a primitive equation model with a turbulent mesoscale eddy field and
uniform wind forcing gave a similar conclusion (Danioux et al., 2008). Non-linear interactions
also provide a mechanism for increasing the vertical wave number, thus for larger vertical shear
and the dissipation and reducing the decay time scale (Davies and Xing, 2002; Zedler, 2009). In
addition to modifying the characteristics of the inertial-gravity waves, nonlinear advection related
to the front can transport NIOs away from the storm track and to higher latitudes (Zhai et al., 2004).
Recent observations and numerical studies in Gulf Stream, Kuroshio, and Japan Sea revealed the
role of vertical circulation on the generation and radiation of near inertial energy (Whitt and
Thomas, 2013; Nagai et al., 2015; Rocha et al., 2018; Thomas, 2019).

The South China Sea (SCS) is a region with frequent tropical cyclone occurrence, ~10.3

each year (Wang et al., 2007). Observations (Sun et al., 2011a,b; Xu et al., 2013) and numerical
studies (Chu et al., 2000) indicated that these TC events are sources for near-inertial energy bursts.
Additionally, the SCS circulation contains  abundant energetic flow (e.g. Qu, 2000; Gan et al.,
2006; Gan et al., 2016a) and mesoscale features, such as a strong coastal jet off the Vietnamese
coast (e.g. Gan and Qu, 2008) and eddies (e.g. Chen et al., 2012). The distribution and evolution
of the TC-induced NIOs are susceptible to the influence of these background currents. (Sun et al.,
2011a; Sun et al., 2011b). However, the estimate of the total contribution of a TC to the KEni in
the SCS is difficult to obtain from observations or linear wave theory due to sparse spatial
observations and the non-homogenous nature of SCS circulation.





In this study, we apply a well-validated numerical model with specific China Sea
configurations to examine the response of KEni to a large TC and background stream jet over the
sloping topography in the SCS. A description of Typhoon Neoguri and the details of the numerical
model implementation are given in Section 2. In Section 3, the general characteristic response of
the near-inertial current and the energy fluxes during the TC forced stage and their later relaxation
stage as TC moved away from the concerned region are presented. Following Section 3, the KEni
equation is used to identify the dynamic processes of the vertical viscous dissipation, pressure
work, and nonlinearity during different phases of the NIOs.
**2 Typhoon Neoguri (2008) and the Ocean Model**
**2.1 Typhoon Neoguri (2008)**
Typhoon Neoguri formed east of the Philippines and entered the SCS on April 15. It first moved
west-northwest with an average translation speed of 5.8 m s$^{-1}$ before it slowed down to 1.6 m s$^{-1}$
on April 16, based on the Joint Typhoon Warning Center (JTWC) best track data, (Fig. 1). On
April 17, Neoguri sped up to 3.8 m s$^{-1}$, turned more northward, and developed into a typhoon with
a maximum wind speed 51 m s$^{-1}$ and a MSLP (minimum sea level pressure) at 948 hPa, at 1800
UTC on April 17 near the Xisha Islands. The Neoguri was a supercritical typhoon travelling with
a translation speed greater than the first baroclinic wave speed. After skirting Hainan Island on
April 18, Neoguri moved northward, weakened to a tropical storm, and further dissipated as it
moved farther inland. The NIO burst induced by Neoguri was shown by the clockwise ($A_{cw}$) and
counter-clockwise ($A_{ccw}$) rotary current amplitudes (m s$^{-1}$), from a current meter mooring at
Wenchang station, to the east off Hainan Island (Fig. 1c).



## 2.2 Ocean Model

We use the China Sea Multi-scale Ocean Modeling System (CMOMS) (Gan et al., 2016a; Gan et al., 2016b) in this study. CMOMS is based on the Regional Ocean Modeling System (ROMS) (Shchepetkin and McWilliams,2005), and the model domain covers the northwest Pacific Ocean (NPO) and the entire China Seas (Bohai, Yellow Sea, East China Sea, and SCS) from approximately 0.95ºN, 99ºE in the southwest corner to the northeast corner of the Sea of Japan. The horizontal size of this grid array decreased gradually from ~10 km in the southern part to~7 km in the northern part of the domain. Vertically, we adopted a 30-level stretched generalized terrain-following coordinate (s).

The model was forced with 6-hourly actual wind speeds of typhoon Neoguri obtained from the Cross-Calibrated Multi-Platform (CCMP) dataset, with a horizontal resolution of 0.25° (Atlas et al. (2011), ftp://podaac-ftp.jpl.nasa.gov/allData/ccmp/L3.0/flk). Wind stress is calculated based on the bulk formulation by Fairall et al. (2003). The daily mean air temperature, atmospheric pressure, rainfall/evaporation, radiation, and other meteorological variables from April 15 to April 18, 2008 from the NCEP/NCAR Reanalysis 1 were used to derive the atmospheric heat and fresh fluxes. External forcing of depth-integrated velocities ($U, V$), depth-dependent velocities ($u, v$), temperature, $T$, and salinity, $S$, at the lateral boundaries were obtained from the Ocean General Circulation Model for the Earth Simulator (OFES) (Sasaki et al., 2008). Open boundary conditions from Gan and Allen (2005) were applied at the open boundaries.

The model was spun up from January 1, 2005 with winter initial fields (temperature and salinity) obtained from the last three-year mean fields of a 25-year run that is initialized with the World Ocean Atlas 2005 (WOA05, Locarnini et al., 2006, Antonov et al., 2006) data, and forced by wind stress derived from climatological (averaged from 1988 to 2013) monthly Reanalysis of





10 m Blended Sea Winds released by the National Oceanic and Atmospheric Administration
(https://www.ncdc.noaa.gov/oa/rsad/air-sea/seawinds.htm). The dynamic configuration and
numerical implementation of the CMOMS system are described in detail in Gan et al. (2016a,
2016b).

We have thoroughly validated the CMOMS by comparing simulated results with those

obtained from various measurements and findings in previous studies. In particular, we have
validated the extrinsic forcing of time-dependent, three-dimensional current system in the tropical
NPO, transports through the straits around the periphery of the SCS, and corresponding intrinsic
responses of circulation, hydrography and water masses in the SCS (Gan et al., 2016a). We have
also validated the circulation of CMOMS by providing a consistent physics between the intrinsic
responses of the circulation and extrinsic forcing of flow exchange with adjacent oceans (Gan et
al., 2016b). The model is also validated with available ARGO temperature profiles (not shown),
observed sea surface temperature (SST) and currents from a time-series current meter mooring
during Neoguri, as described below.

Three-dimensional, hourly-mean dynamic, and thermodynamic variables from April 10 to

May 10, 2008 were used to examine the near-inertial oscillations in this study. Because the inertial
period (IP) in the SCS is larger than 32 hours (near 22°N), the error induced by the hourly model
output is <3%.
**3 Model result**
**3.1 Characteristic response to the TC**

The evolution of the response to the TC in the ocean with existence of a coastal jet in the

SCS is presented according to different stages of the TC forcing. During the pre-storm stage (PS),
before Neoguri entered the SCS on April 14, the wind stress was relatively weak (<0.1 Pa). A



prominent jet current separated from the Vietnamese coast flowing eastward near 16°N (Fig. 2a),
and characterized the circulation in the western part of the SCS (Gan and Qu, 2008). The jet formed
negative (positive) geostrophic vorticity ($\zeta_g$) to the south (north), with the minimum (maximum)
Rossby number ($\zeta_g/f$) <-0.2 (>0.2) near 15.8°N (16.8°N). During the forced stage (FS, Fig. 2b)
between April 15 and April 19, the SCS was under the direct influence of Neoguri, the wind forcing
became significantly stronger (>0.1 Pa), the KE near the surface (10 m) intensified significantly
(>500 J m$^{-3}$) to the east of the TC, and the coastal jet was suppressed by southward flow.
Meanwhile, a strong local divergence and upwelling formed in the surface and generated a strong
cooling (~1.5°C) belt along the TC path that lasted for more than a week. The cooling zone radiated
hundreds of kilometer away from the core of the TC. These features were well captured by the
TC-induced temperature difference between April 19 and 14 from both simulated (Fig. 3a) and
observed SST (Fig. 3b) (http://podaac.jpl.nasa.gov/dataset/JPL-L4UHfnd-GLOB-MUR). After
the end of the FS on April 20 (Fig. 2c), the jet returned to its pre-storm intensity and shifted slightly
northward (Fig. 2c) when the TC center approached the coast (Fig. 1a). Afterwards, during the
relaxation stage (RS) after April 20 (Fig. 2d), the wind forcing from the TC decreased to <0.05 Pa.
——Rotary spectra shows that the near-inertial response of surface currents to the TC occurred
near the local inertial frequency ($f$ = 0.028 cph) at station Wenchang (112°E, 19.6°N) during the
model simulation period (April 10 – May 5) (Fig. 4). The clockwise rotary spectra is calculated
by:

$S_{cw} = 1/8(P_{uu} + P_{vv} - 2Q_{uv})$,                                             (1)

where $P_{uu}$, $P_{vv}$ and $Q_{uv}$ are auto- and quadrature-spectra, respectively (Gonella, 1972). This
simulated result is highly consistent with the observations in the lower frequency band. A relatively
large discrepancy between the model output and observations at the higher frequency band could





have been caused by many reasons, such as the lack of mesoscale and sub-mesoscale processes in
the atmospheric forcing field, and not resolving the oceanic subscale processes due to the limitation
of current model resolution.
**3.2. Near-inertial response in the upper ocean**
We adopted the complex demodulation method successfully used in previous NIO studies
(Gonella, 1972; Brink, 1989; Qi et al., 1995) to extract the inertial current signal. The simulated
horizontal currents ($\vec{u}_h$) were analyzed for inertial currents ($\vec{u}_i$). The inertial currents contain
clockwise (*cw*) and counter-clockwise (*ccw*) rotating components:
$$u_i + iv_i = A_{cw}e^{-i(\phi_{cw}+ft)} + A_{ccw}e^{i(\phi_{ccw}+ft)}, \qquad\qquad (2)$$
where $u_i$ and $v_i$ are the eastward and northward inertial currents at 10 m in the mixed layer, *A* and
$\phi$ are the amplitude and phase of the rotary currents, respectively. Subscripts represent the
clockwise (*cw*) and counter-clockwise (*ccw*) rotating direction, and *f* is the local Coriolis
coefficient. To obtain the amplitude and phase, we performed harmonic analysis daily with each
segment over one inertial period (IP). Then the rotary amplitude and phase were calculated
following previous studies (Mooers, 1973, Qi et al., 1995, Jordi and Wang, 2008).

The time evolution of the daily rotary currents during the FS and RS in the surface layer

varied spatially and was related to the intensity and translation speed of the TC. On April 15 during
PS, Neoguri affected mainly the region south of 13°N, with a relatively fast translation speed ($U_h$
$> 3C_1$, Fig. 1b) and weaker intensity ($V_{max}\sim$35 m s$^{-1}$). In most areas, *cw* rotary currents were strong
($A_{cw}$ >0.1 m s$^{-1}$) yet decayed quickly after 3 days (<2 IP) (Fig. 5a), while the magnitudes of *ccw*
currents were very small (Fig. 5b). After April 18, Neoguri moved into the region between 14°N
and 18°N, where it intensified more than 40% but moved slower with $U_h\sim$2$C_1$. Both the *cw* and
*ccw* currents possessed larger intensities than in the southern region. The induced *cw* currents





displayed an obvious rightward bias, where the enhanced inertial currents extended to ~350 km to
the right of the track and to ~<150 km to the left of the track. This extension of horizontal scale
was related to the region with wind stress $|\tau|$>0.25 Pa in Neoguri.

The maxima of the *ccw* component were located to the left of the TC's path where the wind

vector (Fig. 6) rotated in the same direction as the ocean currents presented in Fig. 5. The
connection between the right (left) bias of the *cw* (*ccw*) currents with the rotation direction of the
wind vector is in agreement with the explanation of Price (1981). 2-3 IPs (>6 days) after the direct
forcing, the *cw* currents remained significant (>0.2 m s$^{-1}$) in an area extending from 110°E to
116°E. In contrast, the *ccw* components, which rotated in the direction of the Earth's rotation,
dissipated quickly after the wind forcing stopped within ~1 day. This short duration of the forced
inertial motion is in agreement with previous studies (Jordi and Wang, 2008).

Besides the intensity and duration, we also looked at the frequency shift ($\delta\omega=\omega-f$) and the

horizontal scale of the NIOs. The frequency shift from the local inertial frequency was estimated
from the temporal evolution of the phase of the back rotary current: $\delta\omega=-\partial\phi/\partial t$. In the FS, the
maximum frequency shift occurred near the jet (112°E to 115°E, 15°N to 16°N), where $\delta\omega\approx0.08f$
($\Delta\phi\approx\pi/4, \Delta t=3$ days, $f=4\times10^{-5}$ s$^{-1}$ at 16°N). The horizontal scale was estimated from the spatial
variation of the rotary current by calculating the horizontal wave number in the meridional
direction as $k_y= \partial\phi/\partial y$. The largest wave number $k_y \approx 3.1\times10^{-5}$ rad m$^{-1}$ was also found near
the jet.
**3.3 Characteristic near-inertial energy**
***Response in the upper layer***

We focused on the area between 110-115°E and 13-19°N (box in Fig. 5a), defined as the

forced region, where the strongest NIO was produced during FS of Neoguri. We calculated the





wind-induced near-inertial energy flux (or the wind work) using $\vec{\tau}_i \cdot \vec{u}_i$, where $\vec{\tau}_i$ is the band-passed
near-inertial wind stress and $\vec{u}_i$ is the near-inertial current at the surface (Silverthorne and Toole,
2009). A 4[th] order elliptic band-pass filter (Morozov and Velarde, 2008) was applied to obtain
near-inertial motion with a band ranging from $0.8f$ to $1.2f$, where $f$ is the local Coriolis coefficient.
The time series of domain-averaged $\vec{\tau}_i \cdot \vec{u}_i$ over the forced region reveals that significant energy
input took place during the FS, with the peak value about $68 \times 10^{-3}$ W m$^{-2}$ on April 17 (Fig. 7).
Under this large wind energy input, the area-averaged depth-integrated KEni (or *AKEni* hereafter)
in the upper layer (0-30 m) increased significantly from its pre-storm value to a maximum ~1500
J m$^{-2}$ during the FS, with an increase rate of about $16 \times 10^{-3}$ W m$^{-2}$ (Fig. 7a). Despite the continuous
positive wind energy flux, the *AKEni* in the upper layer plateaued, indicating that a large amount
of the wind energy was either propagating out the forced region or was lost to the lower layers due
to entrainment. The detailed mechanisms are discussed in the following sections. After the peak
of FS, the wind work decreased significantly with small negative value around the end of the FS.
The *AKEni* decreased to one half of its peak value within 2 days (decrease rate was about $7 \times 10^{-3}$
W m$^{-2}$). After that, the wind work was almost negligible, and the decrease rate of *AKEni* became
smaller (~$0.8 \times 10^{-3}$ W m$^{-2}$).
***Response at depths***

During the FS, the *AKEni* in the upper 200 m constituted ~90% of the total *AKEni* in the

whole water column, while the *AKEni* between 30-200 m alone accounted for ~30-50% (Fig. 7b).
The KENI in this mid-layer had a temporal evolution different from that in the upper layer. It
reached its maximum on April 20, around one and a half days later, and was more than 80% of the
peak value of the *AKEni* in the upper layer (~1000 J m$^{-2}$). Compared to the upper layer *AKEni*, the
the mid-layer *AKEni* during FS increased slightly more slowly (~ $4 \times 10^{-3}$ W m$^{-2}$) while from April





20 to May 5 during RS it decreased much more slowly (~0.61×10$^{-3}$ W m$^{-2}$), and the *AKEni* became
greater than that in the upper layer at the end of FS. The *AKEni* below 200 m was much smaller,
but increased continuously from April 17 to 29, with a rate of about 0.62×10$^{-3}$ W m$^{-2}$, which was
comparable to the *AKEni* rate of decrease in the 30-200 m layer. The *AKEni* in this deep layer
became greater than that in the upper layer after April 25 and that in the layer 30-200 m after April
29; the deep layer reached its maximum value >10 days later.

Spatially, two relatively large KEni patches below upper layer were located to north and

south of ~16°N (Fig. 8a). Their horizontal scales, influenced by near-inertial waves, were much
smaller compared with those in the upper layer. The region with relatively large KEni in the layer
between 30-200 m located near the jet currents, with stronger value during FS than during RS (Fig.
8 a,b). These results suggest that the KEni in this layer might have been determined by both vertical
propagation of the near-inertial gravity wave and horizontal advection of KEni of the background
current. Similar horizontal distribution also occurred below 200 m (Fig. 8 c,d). In contrast to the
layer above, the relatively large value during RS on April 30 indicated a downward propagation
of KEni into the deeper layer. Around the saddle zone west of the Xisha Islands, a relatively large
KEni below 200 m aligned with the 1000 m isobath, and might reflect a topographic effect on the
near-inertial wave.
**3.4 Vertical propagation of near-inertial energy**

It is clear that the distribution of the KEni was mainly controlled by the propagation of

near-inertial wave energy both horizontally and vertically as well as by the background jet stream.
In order to understand the KEni distribution in the deeper water inside the forced region and in the
far field, we selected four different locations, marked as A1, A2, C1, and C2 in Fig. 8 (a-d), for
the analysis of the KEni evolution during the FS and RS. Among them, A1 (113°E, 15.7°N) and





A2 (112.5°E, 16.9°N) are on the right side of the TC track, inside the forced region, and situated
about 200 km apart from each other at the northern (A2) and southern (A1) sides of the jet,
respectively. C1 (114.9°E, 16.9°N) and C2 (114.9°E, 18°N) are the corresponding stations in the
far field where relatively strong KEni intensification occurred.
***Forced region (stations A1 and A2)***
*South of the jet stream at station A1*
The time series of the band-passed inertial velocity $u_i$ as a function of depth shows that there was
an upward phase propagation, in which $u_i$, in the layers below 100 m was leading the upper 50 m
(Fig. 9a). Accompanying this phase propagation was a downward propagation of surface KEni,
which was represented by the lowering of the $u_i$ maxima as a typical Poincaré wave (Kundu and
Cohen, 2008). There were two phases of vertical energy propagation: 1) during FS, there was a
rapid extension of the large $u_i$ maxima to below 100 m from April 17 to 20, and 2) during RS, the
center of the large $u_i$ value descended from ~100 m to 280 m from April 25 to May 5. The vertical
propagation velocity, $C_{gz}$, estimated from this downward transport, was ~17 m day$^{-1}$.
During the first phase, the KEni in the top 30 m and in the 30-200 m layer shared a similar
rate of increase on April 17, indicating that the enhancement of KEni at 30-200 m was related to
the entrainment between the upper and deep layer (Fig. 9c). While the KEni in the upper 30 m
decreased quickly from April 18, it kept increasing at depths from 30-200 m, suggesting that other
contributing mechanisms existed besides the entrainment. The KEni below 200 m also experienced
notable intensification, with a smaller increasing rate than that found in the 30-200 m layer (Fig.
9c). Because the viscous effect is small in the deeper water, this enhancement of the KEni was
most likely associated with the propagation of an inertial-gravity wave.
During the second phase, the KEni in the 30-200 m layer decreased significantly at station




A1, indicating the existence of either downward or horizontal energy transport. From the linearized
inertial-gravity wave equation under the influence of background vorticity, $C_{gz}$ can be obtained by
$$C_{gz} = \frac{\omega^2 - f_{eff}^2}{\omega m},$$
(3)

where $\omega \approx 1.08f$ is the frequency with maximum $S_{cw}$ at 200 m (Fig. 9e); $f_{eff}=f+\zeta_g/2$ is the effective
Coriolis coefficient; and $\zeta_g/f=0.16$ at A1. $m$ is the vertical wave number that we chose to be the
first baroclinic mode under a two-layer approximation based on the stratification (blue line in Fig.
10). From Eq. (3), $C_{gz}$ was about 13.6 m day$^{-1}$, which was in the same range as the modeled $C_{gz}$.
Consistent with the case of $(\omega_0-f_{eff})/f_{eff}<0.1$ in Kunze (1985), the background vorticity in our case
accounted for more than 90% of the modification of the magnitude of the wave dispersion property.
Meanwhile, the KEni in the layer below 200 m did not increase notably, suggesting that other
mechanisms besides vertical propagation of the near-inertial gravity wave might have been
important in the evolution of KEni in water deeper than 200 m.
*North of the jet stream at station A2*
At location A2, strong $u_i$ was mainly trapped in the water above 100 m, and below 100 m
$u_i$ <10 cm s$^{-1}$. It returned to its pre-storm magnitude after 5 IPs (Fig. 9b). The local upwelling
related to the positive background vorticity and to the associated strong surface divergence might
have caused the smaller vertical scale of $u_i$ at A2 (Fig. 4b). The KEni was generally smaller than
that at A1, and relatively large energy was found only in the ML (Fig. 9d). $C_{gz}$ at A2, estimated
from Eq. (3), was 1.2 m day$^{-1}$ ($f_{eff}$ =1.065$f$, $\omega \approx 1.1f$ , $f=4.2\times10^{-5}$s$^{-1}$, and $m=2\pi/30$ m), which was
about one tenth of that at A1. This is consistent with the lack of a distinct pattern of vertical
propagation of NIOs at this station, as shown in the band-passed $u_i$ (Fig. 9b), and the presence
(absence) of a near-inertial peak of $S_{cw}$ at 10 m (200 m) (Fig. 9f).
***Far field region (stations C1 and C2)***



C1 and C2 are located ~400 km to the right of the forced region. During the FS, $u_i$ (Fig. 11a,b) and
KEni (Fig. 11c,d) in the upper layer were smaller than those at those stations in the forced region
due to the weaker TC influence. Only a small downward propagation was discerned during the FS
(Fig. 11a,b). However, notable intensification of the KEni occurred in the layers below the upper
layer after April 23. At C1, the $S_{cw}$ at 10 m had a small red shift, while the $S_{cw}$ at 200 and 500 m
displayed blue shifts with peaks near $1.07f$ (Fig. 11e). The difference between $S_{cw}$ in the upper
layer and in the layers below implies another source of KEni other than local inertial-gravity wave
vertical propagation.

At C2, downward energy propagation appeared after April 23, reaching 100 m from the

surface within 7 days, giving $C_{gz} = 14.3$ m day$^{-1}$ (Fig. 11b). Unlike C1, the intensification was
mainly in the 30-200 m layer. The $S_{cw}$ at both 10 m and 200 m had a broad energy band near the
local $f$ (Fig. 12f). Because $\zeta_g/f = 0.12$ and $f_{eff} = 1.06f$, $C_{gz}$ estimated from Eq. (3) had an upward
propagation, which cannot explain the downward propagation here. The linearized wave theory,
with the consideration of Doppler drift due to background currents, does not seem to be valid in
this location. We will discuss this issue in the next section.
**4 KEni Budget**
We utilized the KEni equation to provide a further analysis of the source of KEni in the water
column. Because the horizontal component of near-inertial kinetic energy is significantly larger
than the vertical component (Hebert and Moum, 1994), we used the horizontal component to
represent the KEni. The KEni budget can be obtained from the horizontal momentum equation.
Following Silverthorne and Toole (2009), the energy equation becomes:
$$\underbrace{\frac{\partial KENI}{\partial t}}_{RATE} = \underbrace{-\vec{u}_i \cdot \langle \nabla_H p \rangle}_{PRES} \underbrace{-\rho_0 \vec{u}_i \cdot \langle \vec{u}_h \cdot \nabla_H \vec{u}_h \rangle}_{NLh} \underbrace{-\rho_0 \vec{u}_i \cdot \left\langle w \frac{\partial \vec{u}_h}{\partial z} \right\rangle}_{NLv} \underbrace{-\rho_0 \vec{u}_i \cdot \left\langle \frac{\partial}{\partial z}\left( v \frac{\partial \vec{u}_h}{\partial z} \right) \right\rangle}_{VVISC} \quad (4)$$





where *AKEni* is the area-averaged depth-integrated near-inertial energy; $\vec{u}_i$ and $\vec{u}_h$ are the near
inertial velocity vector and horizontal velocity, respectively; $p$ is pressure; $\rho_0$ is the reference
density; $\nabla_h$ is the horizontal gradient operator; $w$ is the vertical velocity; $\upsilon$ is the viscosity
coefficient; and the angle bracket represents band-passed filtering on the near-inertial band. The
*PRES* term on the right side of equation represents the pressure work on the *AKEni*, which is
associated with the inertial-gravity wave propagation. $NL_h$ and $NL_v$ represent the horizontal and
vertical divergence of energy flux that include the effects of 1) the advection of *AKEni* due to
background currents and 2) the straining of the wave field due to the background shear currents.
Zhai et al. (2004) found that the geostrophic advection of *AKEni* contributed most of the $NL_h$ and
was the main mechanism for transporting the NIOs in the absence of baroclinic dispersion of
inertial-gravity waves. It was also found to be more important than the dispersive processes along
the Gulf Stream or shelf-break jet. *VVISC* is the vertical viscous effect. As before, we integrate the
KEni vertically in three layers: the upper layer (0-30 m), the subsurface layer (30-200 m), and the
deep layer (>200 m). In the following sections, the *AKEni* budget is considered in entire forced
region (Fig. 5) as well as at the specific stations along the jet stream.
**4.1. Mean balance**
Figure 12 shows the time series of the *AKEni* budget over the entire forced region defined in Fig.
5. The time-averaged horizontal distributions of each term are presented in Fig. 13. During the FS,
the increase of *AKEni* in the upper layer was mainly attributed to the wind energy input because
the *VVISC* term was one order larger than the other terms, with a maximum of $30 \times 10^{-3}$ W m$^{-2}$ on
April 17 (Fig. 12a). The time-integrated *VVISC* during the FS was $2.15 \times 10^3$ J m$^{-2}$ (Table 1).
Stronger *VVISC* in the upper 30 m occurred in the region between 14°N and 18°N (Fig. 13a) along
the TC track with a rightward bias, similar to the distribution of current intensity (Fig. 5c). Like



the wind work during the FS (Fig. 7a), *VVISC* became negative after April 19, indicating the *AKEni*
removal by negative wind work. The influence of *VVISC* extended to the 30-200 m layer, and
provided a positive energy flux (~$1\times10^3$ J m$^{-2}$) in this layer (Figs. 12b, 13b). The effect of *VVISC*
in the deep layer was negligible (Fig. 12c).

Shortly (~1 day) after the large injection of KEni into the upper layer during the FS, the

*PRES* became significant (Fig. 12a, Table 1) and its horizontal distribution resembled that of
*VVISC* (Fig. 13a, d), suggesting that *PRES* radiated the KEni out of the forced region. It provided
a negative KEni flux in the upper layer (-$0.65\times10^3$ J m$^{-2}$), which was largely compensated by the
positive flux in deeper layers (Table 1). This suggests that, during the FS, the main role of the
pressure work was to transport the KEni from the upper layer to the deep layers, and <15% of the
KEni was horizontally propagated outside the forced region.

During the RS, the *VVISC* was relatively small in the upper layer and it accounted for one

third of the *AKEni* removal in the layers below (Table 1). The *PRES* became a major sink for
*AKEni* in the ML (-$0.85\times10^3$ J m$^{-2}$) and subsurface layer (-$0.16\times10^3$ J m$^{-2}$), but was the major
source in the water below 200 m. The *AKEni* loss due to the horizontal wave propagation outside
the forced region was ~-$0.42\times10^3$ J m$^{-2}$, accounting for about 40% of the total loss in the whole
water column.

Nonlinear advection terms had an important influence in the top 200 m but made little

contribution to the *AKEni* budget in the water below 200 m (Fig. 12, Table 1). The horizontal
effects of $NL_h$ and $NL_v$ in these layers were mainly limited to a smaller region, as compared to the
*VVISC* and the *PRES;* and their relatively large values occurred near the slope and the jet (Figs.

13).





In the upper layer, $NL_h$ advected the KEni from the source region; $NL_h$ had positive and
negative values on the eastern and western sides of the TC track, respectively (Fig. 13g). Similar
features, but with much weaker amplitude, were found in the layers below (Fig. 13h,i). During the
FS, in the 30-200 m layer, the domain-averaged $NL_h$ was positive ($0.21\times10^3$ J m$^{-2}$), indicating a
possible extraction of the KEni from background flows. $NL_v$ was a strong energy sink in the upper
200 m (~$0.64\times10^3$ J m$^{-2}$). The TC wind field generated a strong surface horizontal divergence and
upwelling around 16-17°N (Fig. 4b). As a result, the smaller KEni in the lower layer was advected
to the surface east of the Xiasha Islands. This lower KEni generated a negative gradient with
ambient water and resulted in the strong eastwards transport of KEni in the eastward jet current.
As a result, a positive $NL_h$ center located around the area with the strongest negative $NL_v$, and a
negative $NL_h$ center lay to the west of the positive maximum of $NL_h$. During the RS, $NL_h$ became
negative for all layers and provided ~1/3 of the total KEni loss in the water column (-$0.35\times10^3$ J
m$^{-2}$), while $NL_v$ over the whole water column was significantly reduced.
**4.2 Role of the jet stream**
We further show the distinct *AKEni* balance in the southern and northern sides of the jet stream.
During the FS, *VVISC* at A1 on the southern side of the jet stream was the dominant *AKEni* source
in both the upper layer and the 30-200 m layer (Fig. 14a,b), consistent with the large vertical scale
on the southern side of the jet due to local negative vorticity. The enhancement of near-inertial
currents in the upper layer and the concurrent current divergence resulted in the vertical oscillation
of isopycnals (pressure) below the upper layer at this station. During this pumping process, the
*AKEni* in the upper layer was partly transported downward by the *PRES* and partly by the $NL_v$.
During the RS, the *PRES* became the main factor in the *AKEni* budget. It changed from source to
sink in the 30-200 m layer, because of less downward KEni flux from the upper layer. In the deeper





layer, the negative *PRES* indicated that there was a near-inertial wave propagating away from this
location. The positive *AKEni* flux provided by $NL_h$ weakened the effect of the negative *PRES*.

During the FS at A2 on the northern side of the jet stream, *VVISC* in the upper layer (Fig.

14d) had slightly larger magnitude than that at A1. However, it greatly decreased to $3\times10^{-3}$ W m$^{-2}$
below the ML (Fig. 14e), which suggested that the smaller vertical scale on the northern side of
the jet limited the deep penetration of the wind energy in this location. Compared to A1, the $NL_v$
was much stronger in the upper layer, and about one half of the lost energy was compensated for
by the $NL_h$. The *PRES* was negligible compared to that at A1 (Fig. 14d-f). During the RS, the *PRES*
in the ML became a notable sink after April 21 and was accompanied by a positive $NL_h$ (Fig. 14d).
This suggests that the strong jet increased the *AJEni* through either advection or wave propagation
due to *PRES* as a result of jet-NIO interaction at this station. In the deeper layer, the *PRES* provided
a positive *AKEni* flux. From the spectral analysis, the wave at 500 m had a large blue shift of
>0.15*f* (Fig. 9f) that cannot be explained by the background vorticity alone. The wave likely
originated from the northern latitude.

In the far field at stations C1 and C2, where there was no direct wind forcing from the TC,

surface forcing (*VVISC*) was relatively small during the FS (Fig. 14g,j). Therefore, horizontal
transport of energy is needed to sustain the KEni intensification at these two locations (Fig. 11c,d).
At C1, which was on the southern side of the jet stream (Fig. 8) and had a negative background
vorticity, the *PRES* was the main source of *AKEni* in both subsurface and deep layers (Fig. 14h,i).
The existences of the blue shift near the local inertial frequency (Fig. 11e) and of the negative
background vorticity suggest the presence of a southward propagating near-inertial wave towards
C1 from northern region. Because C2 lies near the northeastward turning point of the jet (Fig. 8),
the nonlinear effect became significant in the 30-200 m layer where the jet was strongest (Fig.





14k). After the enhancement of *AKEni* in the subsurface layer, the *PRES* further transported the
KEni downwards and became the major source for the increase of *AKEni* in the deep layer after
April 27 (Fig. 14l). The northeast current advected the lower frequency NIO from the lower
latitude towards the higher latitude, C2, which explains the red shift of the NIO at this location
(Fig. 11f).
**5 Summary**

TCs force the ocean to form NIOs. The response of NIOs is largely associated with the

different forcing stages of the TCs and background flow. Due to spatiotemporally limited
measurements, our understanding of the process and mechanism that govern the NIO response is
mainly based on theories that are constrained by idealized assumptions. In this study, we utilize a
well-validated circulation model to investigate the characteristic response of KEni to a moderately
strong TC (Neoguri) with observed strong KEni and to a unique background circulation.

The near-inertial currents in the upper layer strengthened significantly during the TC forced

stage and displayed a clear rightward bias due to stronger wind forcing and the resonance between
the wind and the near-inertial currents. The distribution of near-inertial currents and the associated
rotary spectra showed that the propagation patterns of NIOs varied greatly from location to
location and were closely linked to the influences of the background jet.

We calculated the KEni balance to diagnose spatiotemporally varying responses and

processes of the near-inertial signals in terms of different forcing stages of the TC. Results show
that during the forcing period, the vertical viscous term, which represents the wind work and
entrainment at the base of the upper layer, was the KEni source in the upper layer.  Around 0.5 IP
after the maximum TC forcing, the pressure work became the main KEni sink in the upper layer,
transporting KEni in the ML into the deeper layers through inertial pumping (upwelling). The

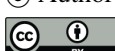



upwelling, caused by the TC-enhanced divergence, also advected smaller KEni from deeper layers
to weaken the KEni in the upper layer.
During the TC relaxation stage, the loss of KEni in the forced region of the whole water
column were caused by the vertical viscous term, the pressure work, and horizontal advection
effects (Table 1). However, these effects acted differently in different layers. The viscous effect
mainly occurred inside the water column, but decreased to near zero in the upper layer after the
direct impact of Neoguri. The pressure work mainly transported the KEni out of the forced region
horizontally and out of the upper layer vertically. It was strongest on the southern side of the jet,
where the negative background vorticity located. The horizontal nonlinear effect also contributed
greatly to the KEni balance near the jet region. It acted as a major sink of KEni by horizontally
advecting the NIO away from the forced region. For locations away from forced region, both near-
inertial wave propagation and horizontal advection contributed to the intensification of the KEni.
We examined the NIOs processes and underlying dynamics in response to different stage
of the TC in the semi-enclosed SCS under influence of unique and strong basin-wide circulation.
Unlike similar study in the SCS, this study enriches our understanding of the spatiotemporal
variability of TC-induced NIOs and provides a useful physical guidance for future  process-
oriented field experiment in the SCS as well as in other subtropical marginal seas that are
frequently affected by the TC.
**Acknowledgments.**
This research was funded by the General Research Fund of Hong Kong Research Grant Council
(GRF16204915) and the Key Research Project of the National Science Foundation of China
(41930539). The buoy data was provided by Qi He from CNOOC Energy Technology & Services



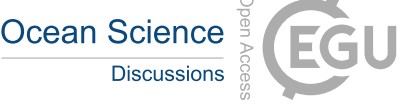

Limited, China. We are also grateful for the support of The National Supercomputing Center of
Tianjin and Guangzhou.



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





**List of Table**

Table 1. Time integrated KEni budget (unit: ×103 J m-2) during the FS and RS.





**List of Figures**

Figure 1 (a) Track of typhoon Neoguri (2008) from JTWC; blue triangles represent ARGO drifter locations and the red circle represents Wenchang where there were ADCP observations. The date is marked by month/day along the track. TS: tropical storm, STS: strong tropical storm, T: typhoon, ST: strong typhoon, and Super T: super typhoon. (b) translation speed ($U_h$, unit: m s$^{-1}$) and the 1$^{st}$ baroclinic wave speed ($C_1$, unit: m s$^{-1}$) along the TC track; (c) clockwise ($A_{cw}$, green line) and counter-clockwise ($A_{ccw}$, blue line) rotary current amplitude (m s$^{-1}$) from current measurement at Wenchang. The *x*-axis is time marked as month/day.

Figure 2 Daily mean KE (J m$^{-3}$, color contour) and current vectors (arrows) at 10 m (a) on April 14 of the pre-storm stage (PS), (b) on April 18 during the strongest wind forcing of the forced stage (FS), (c) on April 20 after the end of the FS, and (d) on April 30 during the relaxation stage (RS). The grey contours are the 200 m, 500 m, and 1000 m isobaths. The magenta line represents TC track. Yellow triangle on April 18 represents the TC location. The TC was located beyond the plotting domain during the other three days, as shown in Figure 1a. The velocity magnitudes<0.2 ms$^{-1}$ are not shown in the vectors.

Figure 3: ΔSST (SSTApril 19-SSTApril14) from (a) model results and (b) GHRSST JPL MUR satellite products. The pink curve refers to the trajectory of the TC Neoguri.

Figure 4 Rotary spectra of clockwise component (upper 10 m) at Wenchang (112°E, 19.6°N) from model simulations (red) and observations (blue).

Figure 5 Time series, represented by color bar, of daily (a) clockwise and (b) counter-clockwise rotary current vectors from April 14 to 30 during different stages of the TC forcing, signifying the response of the current to the local wind rotation. For the clockwise (counter-clockwise)





component, only currents with magnitude larger than 0.2 (0.05) m s$^{-1}$ are shown. The black box represents the forced region.

Figure 6 Time series of 6-hourly wind stress vectors during the forced-stage (FS) from April 15-20.

Figure 7 Time series of (a) the area-averaged wind energy flux into the near-inertial band (unit: $10^{-3}$ W m$^{-2}$) and (b) depth-integrated KEni (J m$^{-2}$) in the forced region for different layers.

Figure 8 Daily averaged KEni (KJ m$^{-2}$) of layers (a, b) 30-200 m and (c,d) below 200 m on (a, c) April 20 during FS, and (b,d) April 30 during RS. The thick red arrows show the location of the jet stream (Fig. 2), while the orange curve arrows indicate regions with relative vorticity $\zeta > 0$, and the blue curve arrows indicate regions with $\zeta < 0$ induced by the jet. Stations A1 and A2 are on the right side of the TC track at the northern (A2) and southern (A1) sides of the jet, respectively. Stations C1 and C2 are corresponding stations in the far field. Station B is located in the upstream of the jet stream.

Figure 9 Time series of (a, b) $u_i$ (m s$^{-1}$), (c, d) KEni (J m$^{-2}$), and (e, f) rotary spectra (cw component) at locations A1 (a,c,e) and A2 (b,d,f).

Figure 10 Time-averaged $N^2$ (s$^{-2}$) from April 15 to May 5 at locations A1 (red) and A2 (blue).

Figure 11 As in Fig. 9, except for locations C1 (a, c, e) and C2 (b, d, f).

Figure 12 Time series of area-averaged, depth-integrated KEni budget for (a) 0-30 m, (b) 30-200 m, and (c) >200 m in the forced region. Terms represent (unit: $\times 10^{-3}$ W m$^{-2}$): (a-c) vertical viscous effect (VVISC), (d-f) divergence of energy flux (PRES), (g-i) horizontal non-linear interaction (NL$_h$), and (j-l) vertical non-linear interaction (NL$_v$). The vertical lines separate the pre-storm stage, FS and RS during the TC forcing.





Figure 13 Horizontal distribution of time-averaged (April 15-May 5) depth-integrated KEni budget

in different layers: 0-30 m (left column), 30-200 m (middle), and >200 m (right). The terms re

Figure 14 Time series of KEni budget at locations: A1 (a-c), A2 (d-f), C1 (g-i), and C2 (j-l) in

layers: 0-30 m (left column), 30-200 m (middle column), and >200 m (right column).

presented are (unit: $\times 10^{-3}$ W m$^{-2}$): (a-c) VVISC, (d-f) PRES, (g-i) NL$_h$, and (j-l) NL$_v$.





| TERM | RATE | | VVISC | | PRES | | NLh | | NLv | |
|---|---|---|---|---|---|---|---|---|---|---|
| Phase | FS | RS | FS | RS | FS | RS | FS | RS | FS | RS |
| 0-30 m | 1.15 | -0.97 | 2.15 | 0.00 | -0.65 | -0.85 | -0.05 | -0.10 | -0.29 | -0.01 |
| 30-200m | 1.11 | -0.44 | 0.96 | -0.19 | 0.30 | -0.16 | 0.21 | -0.18 | -0.35 | 0.10 |
| >200 m | 0.23 | 0.36 | -0.02 | -0.09 | 0.26 | 0.59 | 0.00 | -0.07 | -0.01 | -0.07 |
| Column | 2.51 | -1.02 | 3.10 | -0.27 | -0.09 | -0.42 | 0.16 | -0.35 | -0.66 | 0.02 |

Table 1. Time integrated KEni budget (unit: $\times 10^3$ J m$^{-2}$) during the FS and RS.



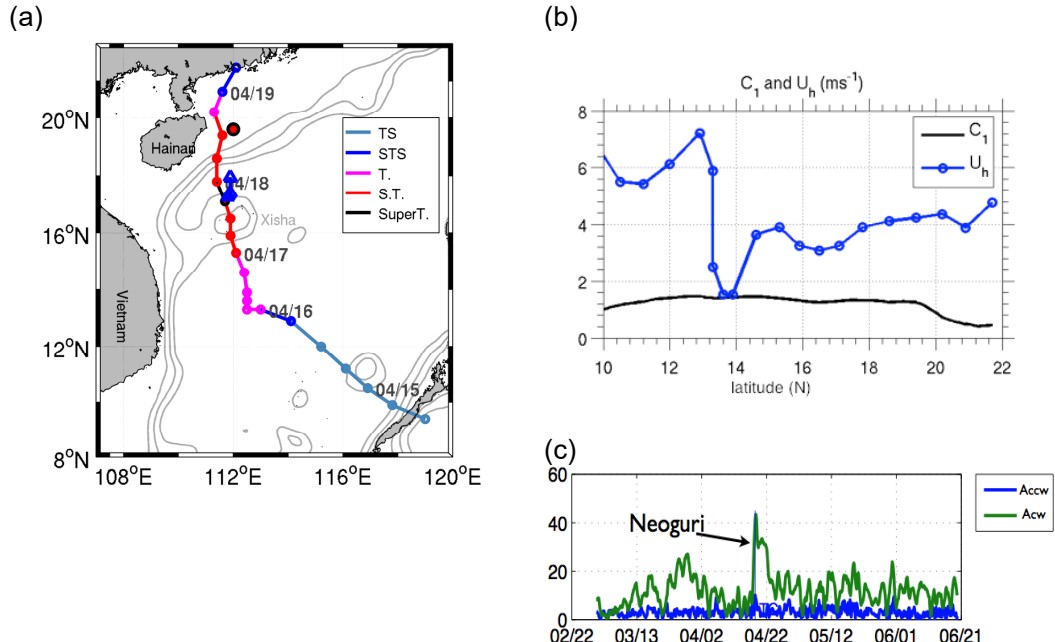

Figure 1 (a) Track of typhoon Neoguri (2008) from JTWC; blue triangles represent ARGO drifter locations and the red circle represents Wenchang where there were ADCP observations. The date is marked by month/day along the track. TS: tropical storm, STS: strong tropical storm, T: typhoon, ST: strong typhoon, and Super T: super typhoon. (b) translation speed ($U_h$, unit: m s$^{-1}$) and the 1$^{st}$ baroclinic wave speed ($C_1$, unit: m s$^{-1}$) along the TC track; (c) clockwise ($A_{cw}$, green line) and counter-clockwise ($A_{ccw}$, blue line) rotary current amplitude (m s$^{-1}$) from current measurement at Wenchang. The $x$-axis is time marked as month/day.


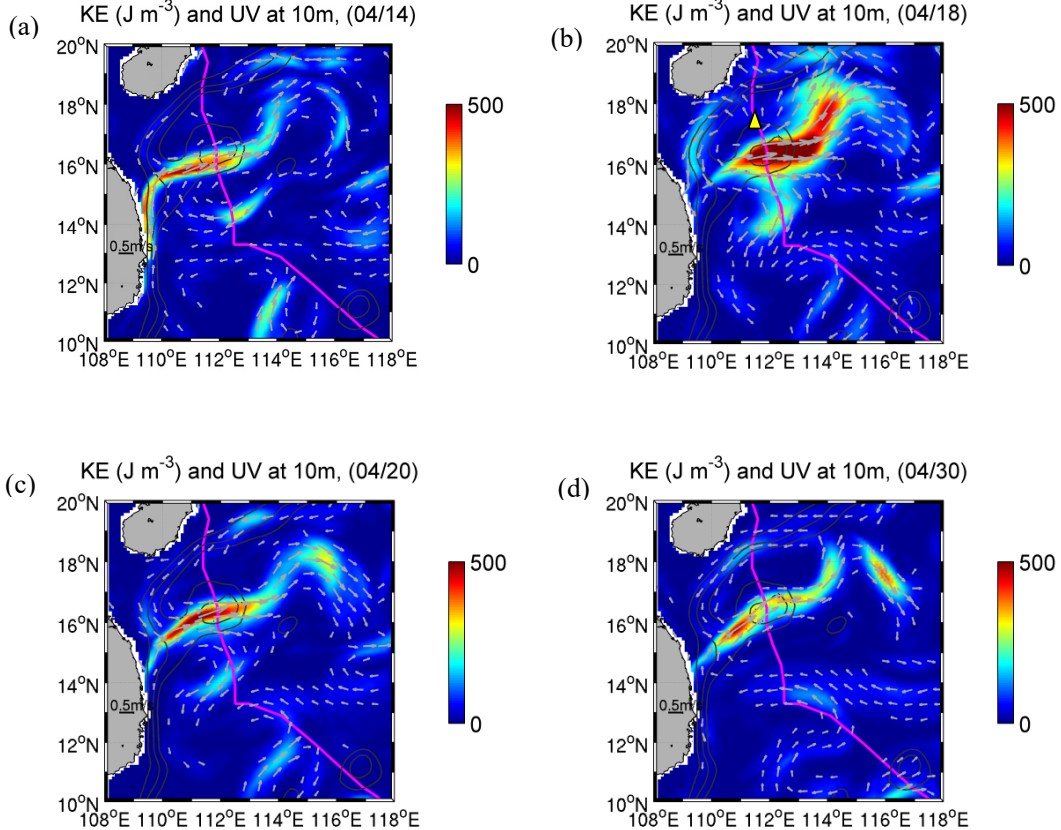

Figure 2 Daily mean KE (J m$^{-3}$, color contour) and current vectors (arrows) at 10 m (a) on April 14 of the pre-storm stage (PS), (b) on April 18 during the strongest wind forcing of the forced stage (FS), (c) on April 20 after the end of the FS, and (d) on April 30 during the relaxation stage (RS). The grey contours are the 200 m, 500 m, and 1000 m isobaths. The magenta line represents TC track. Yellow triangle on April 18 represents the TC location. The TC was located beyond the plotting domain during the other three days, as shown in Figure 1a. The velocity magnitudes<0.2 ms$^{-1}$ are not shown in the vectors.





(a)                                    (b)

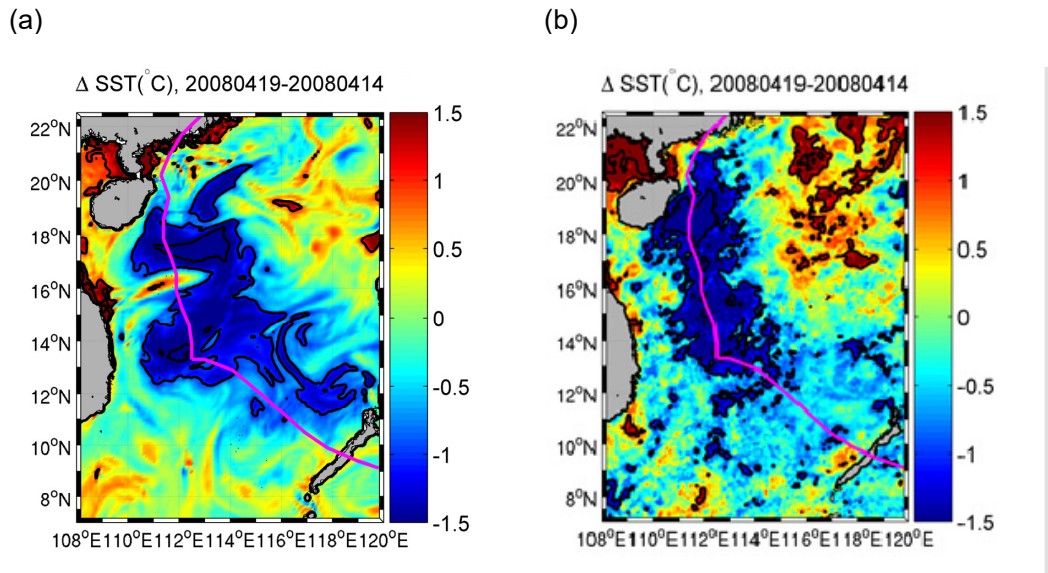

Figure 3 ΔSST (SST$_{April 19}$-SST$_{April14}$) from (a) model results and (b) GHRSST JPL MUR satellite products. The pink curve refers to the trajectory of the TC Neoguri.



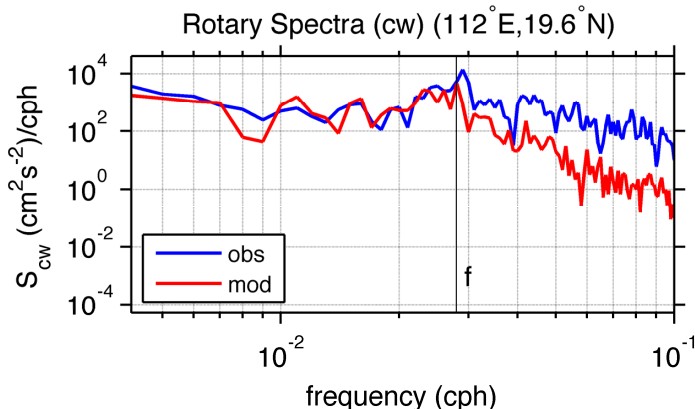

Figure 4 Rotary spectra of clockwise component (upper 10 m) at Wenchang (112°E, 19.6°N)
from model simulations (red) and observations (blue).





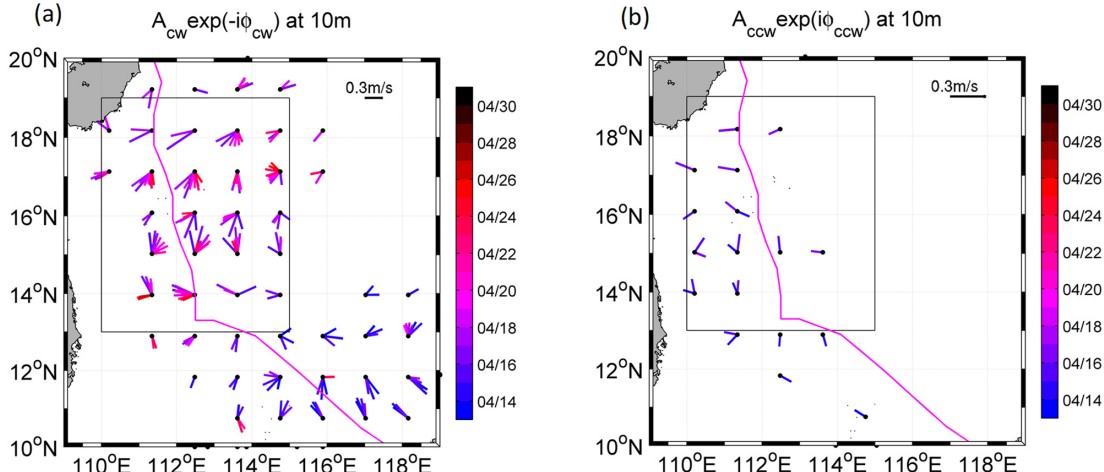

Figure 5 Time series, represented by color bar, of daily (a) clockwise and (b) counter-clockwise rotary current vectors from April 14 to 30 during different stages of the TC forcing, signifying the response of the current to the local wind rotation. For the clockwise (counter-clockwise) component, only currents with magnitude larger than 0.2 (0.05) m s$^{-1}$ are shown. The black box represents the forced region.

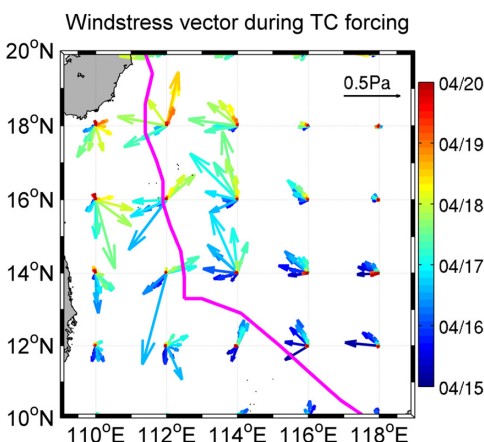

Figure 6 Time series of 6-hourly wind stress vectors during the forced-stage (FS) from April 15-
20.





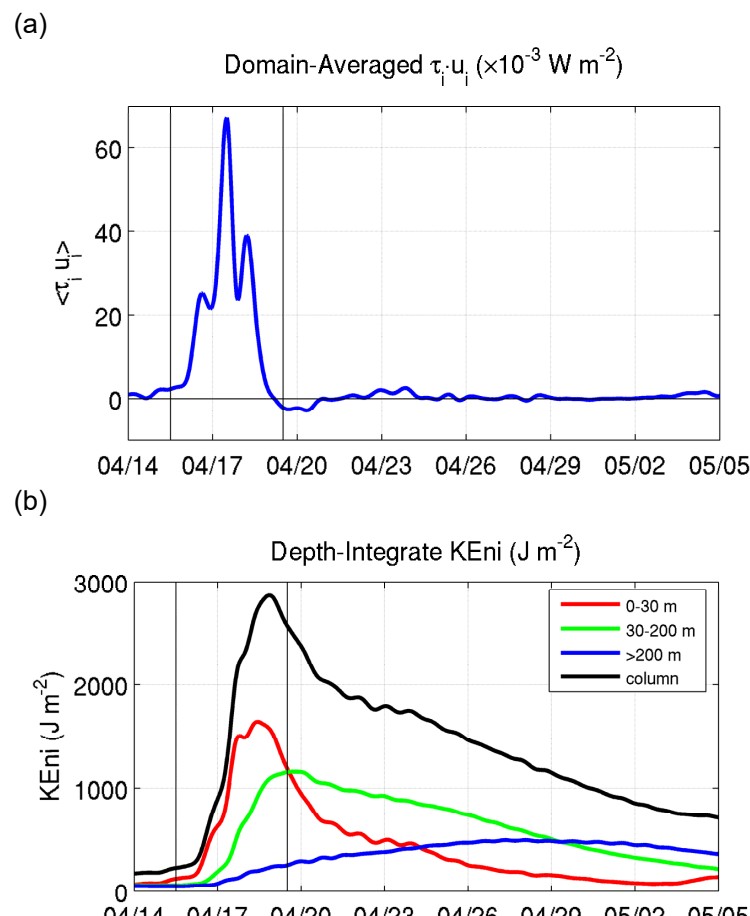

Figure 7 Time series of (a) the area-averaged wind energy flux into the near-inertial band (unit: $10^{-3}$ W m$^{-2}$) and (b) depth-integrated KEni (J m$^{-2}$) in the forced region for different layers.

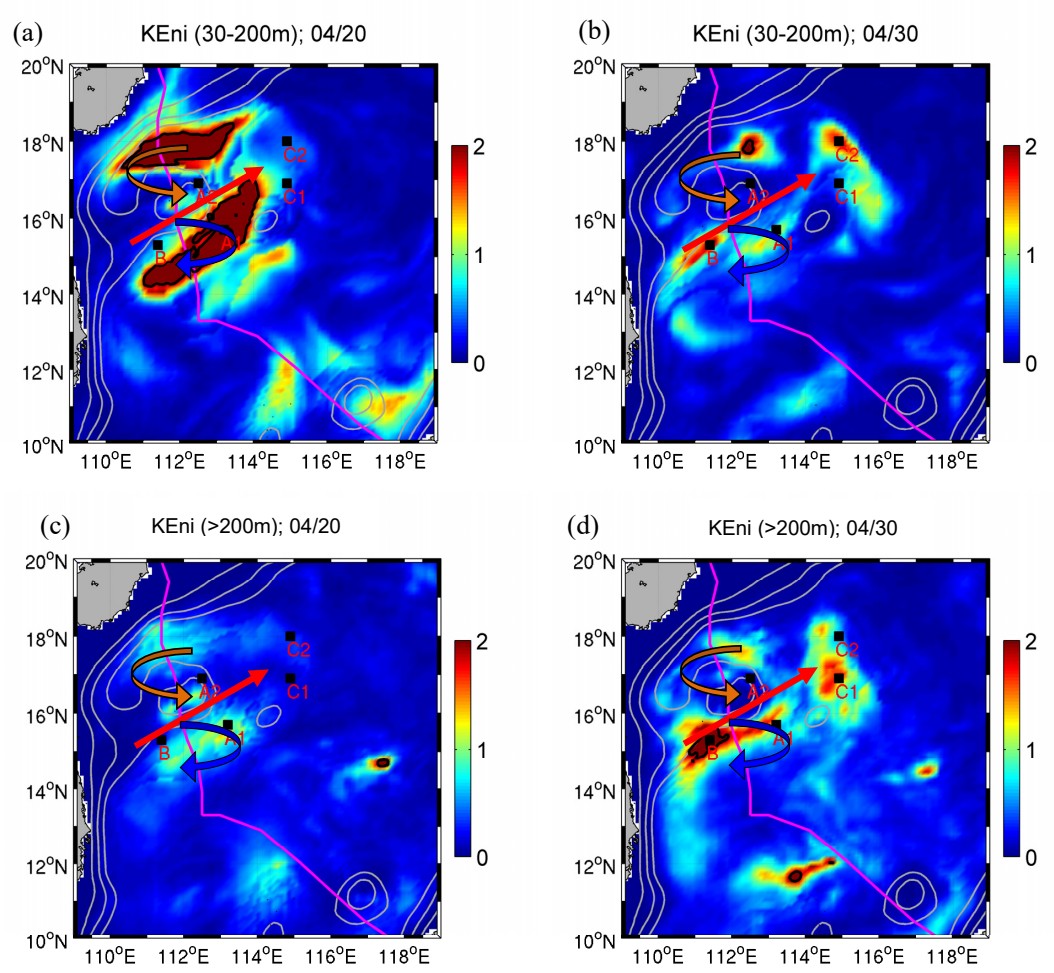

Figure 8 Daily averaged KEni (KJ m⁻²) of layers (a, b) 30-200 m and (c,d) below 200 m on (a, c) April 20 during FS, and (b,d) April 30 during RS. The thick red arrows show the location of the jet stream (Fig. 2), while the orange curve arrows indicate regions with relative vorticity $\zeta > 0$, and the blue curve arrows indicate regions with $\zeta < 0$ induced by the jet. Stations A1 and A2 are on the right side of the TC track at the northern (A2) and southern (A1) sides of the jet, respectively. Stations C1 and C2 are corresponding stations in the far field. Station B is located in the upstream of the jet stream.



Figure 9 Time series of (a, b) $u_i$ (m s$^{-1}$), (c, d) KEni (J m$^{-2}$), and (e, f) rotary spectra (*cw* component) at locations A1 (a,c,e) and A2 (b,d,f).



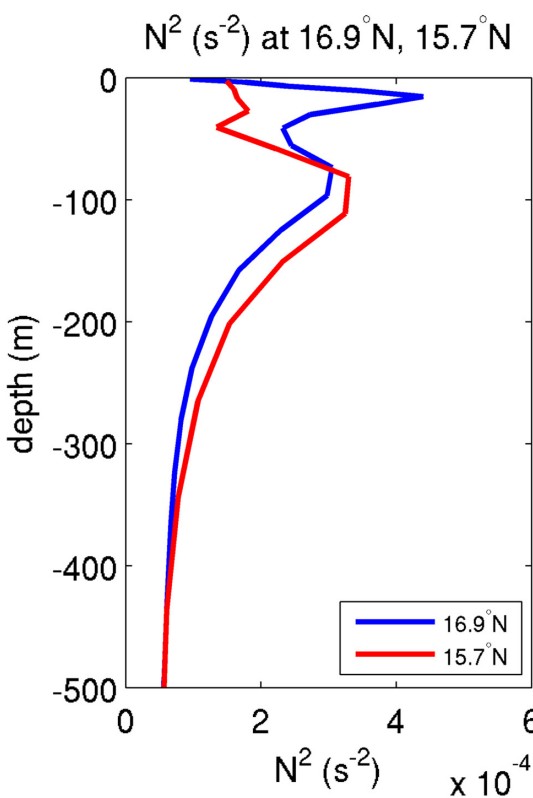

Figure 10 Time-averaged $N^2$ ($s^{-2}$) from April 15 to May 5 at locations A1 (red) and A2 (blue).



Figure 11 As in Fig. 9, except for locations C1 (a, c, e) and C2 (b, d, f).


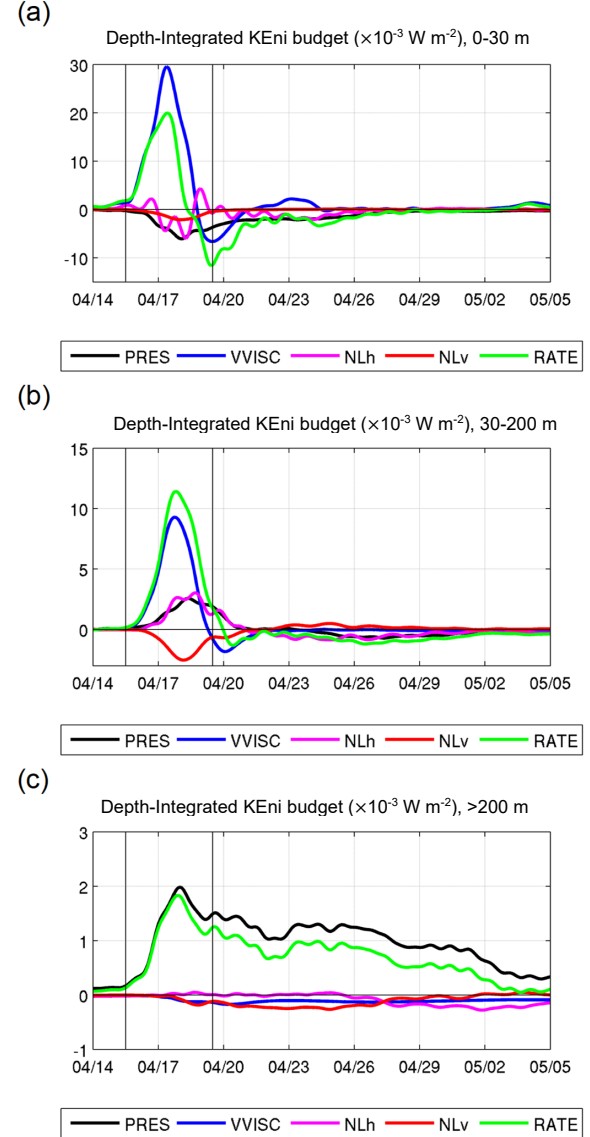

Figure 12 Time series of area-averaged, depth-integrated KEni budget for (a) 0-30 m, (b) 30-200
m, and (c) >200 m in the forced region. Terms represent (unit: $\times 10^{-3}$ W m$^{-2}$): (a-c) vertical
viscous effect (VVISC), (d-f) divergence of energy flux (PRES), (g-i) horizontal non-linear
interaction (NL$_h$), and (j-l) vertical non-linear interaction (NL$_v$). The vertical lines separate the
pre-storm stage, FS and RS during the TC forcing.



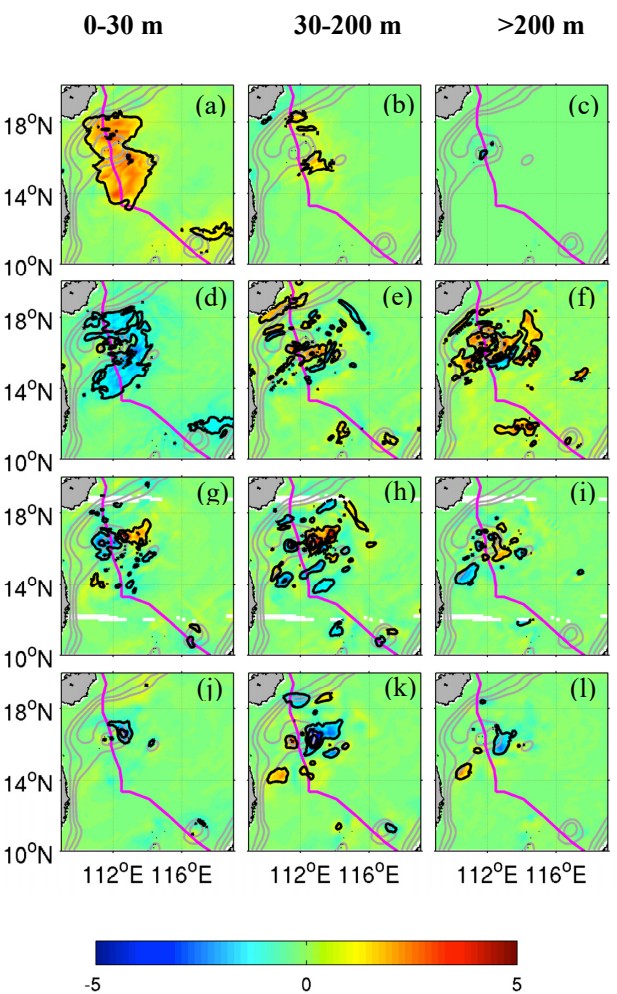

Figure 13 Horizontal distribution of time-averaged (April 15-May 5) depth-integrated KEni budget in different layers: 0-30 m (left column), 30-200 m (middle), and >200 m (right). The terms represented are (unit: ×10⁻³ W m⁻²): (a-c) *VVISC*, (d-f) *PRES*, (g-i) *NLₕ*, and (j-l) *NLᵥ*.




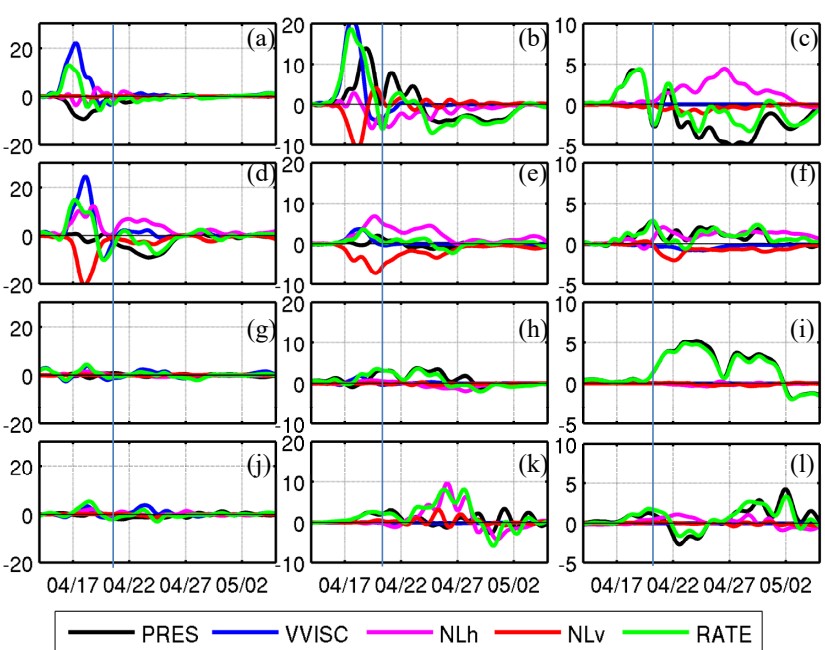

Figure 14 Time series of KEni budget at locations: A1 (a-c), A2 (d-f), C1 (g-i), and C2 (j-l) in layers: 0-30 m (left column), 30-200 m (middle column), and >200 m (right column).