# Peer review of "Response of near-inertial energy to a supercritical tropical cyclone and jet stream in the South China Sea: modeling study"

_Ocean Science, 2020_

## Referee Comment (RC1) · Anonymous Referee #1 · 8 May 2020

The near-inertial motion associated with typhoons' passage is an important topic in the South China Sea. Most of previous research in this region are based on in-situ data. This manuscript provides a modelling study on the spatial and temporal distribution of near-inertial energy. It found an interesting modulation of the near-inertial energy by a jet stream in the South China Sea, with strong (weak) activities at places of positive (negative) vorticity (A1 and A2), and with large values at places even ∼400 km away from the cyclone track (C1 and C2). The investigation of energy budget provides valuable insights into functions of different terms (pressure work, viscous effect, and nonlinear terms) at different stages (forcing and relaxation) in different layers (upper 30, 30-200, below 200). This work merits publishing, however, some questions as follow

should be considered.

Specific comments:

1. The 6-hourly wind from CCMP is used to calculate the wind stress. Since the cyclone induces a wind stress changing rapidly with time, an interpolation from such a long time interval is unreliable and probably underestimates the KEin (Jing, Wu and Ma, 2015, JAOT). A time interval less than 1 hour may be necessary. Furthermore, the wind speed of a typhoon is probably underestimated in the CCMP data. Most of previous research reconstruct the cyclone wind from analytic expressions, such as Holland (1980).

2. The model employed has been well validated and used in several previous research, however, the process of a typhoon response is of short time scale, baroclinic, and intermittent. A validation with ADCP current data at some places is critical.

3. The much weaker KEin at A2 is considered to be due to the positive vorticity induced by the jet. Information on the horizontal and vertical scale of the background vorticity may be necessary. And usually the wind is the first order factor of near-inertial intensity. A comparison of wind time series between A1 and A2 makes sense.

4. The depth of 30 m is used as a boundary between upper and mid-depth layers. There is no clarification why 30 m is chosen, not 50 m or other values.

Technical comments:

1. Figure 1. The value of isobaths should be noted since you mention it in the text (Line 240).

2. Figure 5. The way to display different magnitudes is not good. Think about a better way.

3. Line 221 When the mid-layer and upper layer are firstly mentioned, the exact depth range should be noted. The word 'upper' seems to represent a range much larger than

30 m. Maybe 'surface' or 'top' is a bit more appreciate.

4. Lines 273 and 315: Equations are not clearly seen.

5. Line 393: 'AJEin' may be corrected to 'AKEin'.

---

## Referee Comment (RC2) · Anonymous Referee #2 · 24 May 2020

In this study, near-inertial oscillations (NIOs) generated by a tropical cyclone (TC) and modulated by a background jet stream are simulated. The proposed work is meaningful, and the results are encouraging. However, a few points need to be clarified:

1. Line 61: the transport of KEni is modulated by the velocity and vorticity of a jet stream. This jet stream plays an important role in this study and more descriptions and explanations on the stream is necessary. One reference (Gan and Qu, 2008) has been cited by the author on Line 61, but no further description on the jet. Under what condition will the jet stream exist? Is the intensity/velocity of the stream changes a lot?

The jet stream is lumped with the NIO generated by the TC, and this interaction is

nonlinear. The intensity/velocity of the stream will significantly affect the interaction. Is the stream located in its typical location with its typical velocity during the typhoon season? If a reader wants to know how much KEni will be generated and how the NIO propagates, which is the main topic of the present study, he would want to know whether the scenario used in this study is applicable.

2. Line 98: how the TC forcing is exerted on the sea surface is not clearly explained. CCMP provides the wind speed at 10 m high, can it represent the drag direction on the sea surface?

3. Line 99: Fairall et al. (2003) is not listed in the References.

4. Line 114 : The author emphasizes the model is well-validated, but these validations are in a general sense. Is there any evidence that NIOs can be properly simulated by this model? Some validations should be presented in the paper. For example, the near inertia component of the model can be compared with that of the ADCP data measured by the station shown in Fig. 1.

5. Line 273: eq. (3) is out of nowhere. Please provide references for eq. (3).

---

## Author Comment (AC2) · 3 Jun 2020

In this study, near-inertial oscillations (NIOs) generated by a tropical cyclone (TC) and modulated by a background jet stream are simulated. The proposed work is meaningful, and the results are encouraging. However, a few points need to be clarified:

Response: We appreciate the detailed reading and helpful comments from the reviewer, which are now integrated into the revised manuscript.

Comments:

1. Line 61: the transport of KEni is modulated by the velocity and vorticity of a jet

stream. This jet stream plays an important role in this study and more descriptions and explanations on the stream is necessary. One reference (Gan and Qu, 2008) has been cited by the author on Line 61, but no further description on the jet. Under what condition will the jet stream exist? Is the intensity/velocity of the stream changes a lot? The jet stream is lumped with the NIO generated by the TC, and this interaction is nonlinear. The intensity/velocity of the stream will significantly affect the interaction. Is the stream located in its typical location with its typical velocity during the typhoon season? If a reader wants to know how much KEni will be generated and how the NIO propagates, which is the main topic of the present study, he would want to know whether the scenario used in this study is applicable.

Response: The jet stream was resulting from the (summer) monsoon-driven strong coastal current over narrow shelf topography off Vietnam and it persisted as a distinct circulation feature in the SCS during the summer. The northward flowing coastal current separated from the coast and overshoots northeastward into the SCS basin as it encountered the coastal promontory in the central Vietnam. We now add this information in section 3.1 to provide background for the jet stream shown in Fig. 2

2. Line 98: how the TC forcing is exerted on the sea surface is not clearly explained. CCMP provides the wind speed at 10 m high, can it represent the drag direction on the sea surface?

Response: In ROMS, surface wind stress is calculated from wind speed at 10 m high by using bulk algorithm based on Monin–Obukhov similarity theory, described in Fairall et al. (2003). We have mentioned this in the 2nd paragraph of section 2.2 Ocean Model.

3. Line 99: Fairall et al. (2003) is not listed in the References.

Response: Reference has been added in the list according to the comment.

4. Line 114 : The author emphasizes the model is well-validated, but these validations

are in a general sense. Is there any evidence that NIOs can be properly simulated by this model? Some validations should be presented in the paper. For example, the near inertia component of the model can be compared with that of the ADCP data measured by the station shown in Fig. 1.

Response: The rigorous model validation of circulation and physics have been conducted by Gan et al. (2016a, b), which provides a level confidence for the ocean circulation (e.g. jet stream) and dynamics hub for this study.

The full-scale model-observation comparison is built on both availability of field measurement and advanced theoretical study. We compared the TC induced surface cooling from SST data (Fig. 3), and the rotary energy spectra from ADCP data (Fig. 4) in section 3.1. TC-induced surface cooling is reasonable in both intensity and the spatial coverage. Rotary spectra shows that the model can capture the inertial signal and simulate the low frequency current with reasonable intensity. In addition, we found that the correlation coefficients of near-inertial band-passed velocity between ADCP and model simulation at Wenchang station were 0.62 and 0.57 for east-west (u) and north-south (v) component, respectively, which indicated that the model captured reasonably well the NIOs under the influence of the background circulation of the SCS.

There existed inevitably the model-observation discrepancies, such as differences of velocity magnitude ($\sim$0.06 m s-1) at near-inertial band and rotary spectra at the higher frequency (Fig. 4). These discrepancies could have been caused by many reasons, such as the lack of mesoscale and sub-mesoscale processes in the atmospheric forcing field, the linear interpolation process of the atmospheric forcing (Jing et al., 2015), and not resolving the oceanic subscale processes by the current model resolution. However, these discrepancies will not undermine the discussion about the process and mechanism of near-inertial energy response to the TC and jet stream in this study.

Some of the above information are now integrated into the revised ms..

5. Line 273: eq. (3) is out of nowhere. Please provide references for eq. (3).

Response: citation of Eq. (3) has been added in the reference list.

**[OSD](OSD)**

Interactive
comment

---

## Author Comment (AC1)

*The near-inertial motion associated with typhoons' passage is an important topic in the South China Sea. Most of previous research in this region are based on in-situ data. This manuscript provides a modelling study on the spatial and temporal distribution of near-inertial energy. It found an interesting modulation of the near-inertial energy by a jet stream in the South China Sea, with strong (weak) activities at places of positive (negative) vorticity (A1 and A2), and with large values at places even _400 km away from the cyclone track (C1 and C2). The investigation of energy budget provides valuable insights into functions of different terms (pressure work, viscous effect, and nonlinear terms) at different stages (forcing and relaxation) in different layers (upper 30, 30-200, below 200). This work merits publishing, however, some questions as follow should be considered.*

**Response:** We appreciate the detailed reading and helpful comments from the reviewer, which are now integrated into the revised manuscript.

**Specific comments:**

*1. The 6-hourly wind from CCMP is used to calculate the wind stress. Since the cyclone induces a wind stress changing rapidly with time, an interpolation from such a long time interval is unreliable and probably underestimates the KEin (Jing, Wu and Ma, 2015, JAOT). A time interval less than 1 hour may be necessary. Furthermore, the wind speed of a typhoon is probably underestimated in the CCMP data. Most of previous research reconstruct the cyclone wind from analytic expressions, such as Holland (1980).*

**Response:** We agree that higher-frequency wind forcing and proper temporal and spatial interpolation methods may be required. Besides the availability of higher-frequency wind forcing data, it is not clear how numerical disadvantage of using realistic higher-frequency forcing may affect on resolving the TC-induced NIOs in numerical simulation. In addition, this study is a 'direct simulation" of TC induced NIOs in the South China Sea and analytic expressions TC may not be suitable. We have mentioned this in the revised paper (3.1).

*2. The model employed has been well validated and used in several previous research, however, the process of a typhoon response is of short time scale, baroclinic, and intermittent. A validation with ADCP current data at some places is critical.*

**Response:** The rigorous model validation of circulation and physics have been conducted by Gan et al. (2016a, b), which provides a level confidence for the ocean circulation (e.g. jet stream) and dynamics hub for this study.

The full-scale model-observation comparison is built on both availability of field measurement and advanced theoretical study. We compared the TC induced surface cooling from SST data (Fig. 3), and the rotary energy spectra from ADCP data (Fig. 4) in section 3.1. TC-induced surface cooling is reasonable in both intensity and the spatial coverage. Rotary spectra shows that the model can capture the inertial signal and simulate the low frequency current with reasonable intensity. In addition, we found that the correlation coefficients of near-inertial band-passed velocity between ADCP and

model simulation at Wenchang station were 0.62 and 0.57 for east-west ($u$) and north-south ($v$) component, respectively, which indicated that the model captured reasonably well the NIOs under the influence of the background circulation of the SCS.

There existed inevitably the model-observation discrepancies, such as differences of velocity magnitude ($\sim$0.06 m s$^{-1}$) at near-inertial band and rotary spectra at the higher frequency (Fig. 4). These discrepancies could have been caused by many reasons, such as the lack of mesoscale and sub-mesoscale processes in the atmospheric forcing field, the linear interpolation process of the atmospheric forcing (Jing et al., 2015), and not resolving the oceanic subscale processes by the current model resolution. However, these discrepancies will not undermine the discussion about the process and mechanism of near-inertial energy response to the TC and jet stream in this study.

Some of the above information are now integrated into the revised ms..

*3. The much weaker KEin at A2 is considered to be due to the positive vorticity induced by the jet. Information on the horizontal and vertical scale of the background vorticity may be necessary. And usually the wind is the first order factor of near-inertial intensity. A comparison of wind time series between A1 and A2 makes sense.*

**Response:** We include the vertical profile of the low-passed (3-day) vorticity at A1 and A2 in the revised Figure 10, which represents the vertical scale of the background vorticity. It shows clearly that the vertical scale of the KEni propagation is closely related to the vertical scale of vorticity.

[Figure]

Figure 10 Time-averaged (a) $N^2$ (s$^{-2}$) and (b) low-passed (3 day) vorticity from April 15 to May 5

at locations A1 (red) and A2 (blue).

*4. The depth of 30 m is used as a boundary between upper and mid-depth layers. There is no clarification why 30 m is chosen, not 50 m or other values.*

**Response:** We choose the 30 m as the maximum of domain-averaged $N^2$ (buoyancy frequency) over the forced region located close to 30 m as Figure R1.

[Figure]

Figure R1. Vertical profile of domain averaged (forced region) buoyancy frequency N2 (s$^{-2}$) averaged from Apr. 10 to May 10.

**Technical comments.**
*1. Figure 1. The value of isobaths should be noted since you mention it in the text (Line 240).*

**Response:** We revised Figure 1 accordingly.

*2. Figure 5. The way to display different magnitudes is not good. Think about a better way.*

**Response:** Figure 5 intends to show both spatial and temporal information of rotary current vectors.

*3. Line 221 When the mid-layer and upper layer are firstly mentioned, the exact depth range should be noted. The word 'upper' seems to represent a range much larger than 30 m. Maybe 'surface' or 'top' is a bit more appreciate.*

**Response:** We modified wording accordingly.

*4. Lines 273 and 315: Equations are not clearly seen.*

**Response:** We modified the equation accordingly.

*5. Line 393: 'AJEin' may be corrected to 'AKEin'.*

**Response:** We modified wording accordingly.

---

## Author Response (AR2)

**Response**

*Referee 1 point 4; choice of 30m for upper-layer depth. You refer to the maximum of N^2, but that is very weak and larger values occur not far below. Also figure 10 shows two very different profiles. I wonder if currents' vertical structure or depth-coherence might give a reason for the choice?*

**Response**: We plot out the domain averaged relatively vorticity profile to show the vertical structure of the current field. It shows that, a minimum vorticity value appears near 30 m. Therefore, we choose the 30 m based on the combination consideration of stratification and vorticity fields. We clarified this in the paper.

[Figure]

*Title, lines 4, 6, 16, 17, 67, 134 (and others?) Better to omit "stream". "jet stream" is used for an atmospheric phenomenon.*

**Response**: Modified as suggested.

Line 46. Better "and proven analytically by . ."

**Response**: Modified as suggested.

Line 50.  Better "and dissipation and reduced decay time . ."

**Response**: Modified as suggested.

Line 56.  "~10.3" -> "~10"

**Response**: Modified as suggested.

Line 101 (106).  "fresh" -> "fresh-water"

**Response**: Modified as suggested.

Lines 234-235.  "more than 80% of the peak value of the AKEni in the upper layer (~1000 J m^-2)".  No, neither figure 7b or the values in the text (1500, 1000) show "more than 80%".  "(~1000 . . )" needs moving to follow immediately after what it refers to.

**Response**: Corrected the number according to figure 7b.

Line 242.  "the deep layer reached its maximum value >10 days later".  "later" than what?  Probably this text can be omitted, c.f. line 239.

**Response**: Corrected as "~10 days after the ending of the FS "

Line 318.  "12f" -> "11f"?

**Response**: Corrected the figure number.

Lines 329 and 333, 335, 337.  You already defined AKEni and you don't need it here; (4) is not depth-integrated.

**Response**: Equation (4) is not depth-integrated equation, we did depth-integration and area-averaging based on this equation. Made clarification in the text.

Line 442. "transporting KEni in the ML into the deeper layers through inertial pumping (upwelling)." Why "upwelling" here when you are discussing transport to deeper?

**Response**: Removed the upwelling in the bracket, as the upwelling is mainly discussed in the next sentence.

Line 443. "upwelling, caused by the TC-enhanced divergence" is OK, but omit "The" at the end of line 442.

**Response**: Corrected.

Figures 1, 10, 13, 14. The captions in the "List of Figures" and under the figure are different. [In the list, the last line of figure 13 caption is misplaced and appears after figure 14 caption]

**Response**: Corrected.

Figure 8. "A1", "A2", "C1", "C2" might be written in black for visibility and consistency with the dots.

**Response**: Corrected as comment.

Mark-up manuscript version

[revised manuscript text omitted]

---

## Author Response (AR3)

**Response to Comments from Editor's comment and editing**

*Line 21. ". . frequency," (add ",")*

Done

*Lines 42-43. At present this seems to say that the β-effect reduces the NIO decay time scale. I think you want "the β-effect leads to equatorward propagation of the NIOs and their decay time scale is reduced because of the vertical propagation . . ."*

Done.

*Line 50. Delete redundant "the dissipation and"*

Done

*Line 52. ". . related to a front . ." [because a front was not mentioned previously]*

Done.

*Line 68. Omit "stream"? Maybe you just want ". . background flow over the"*

Done.

*Line 81. ". . pressure) 948 hPa . ." (Omit first "at").*

 Done

*Line 158. "spectra" -> "spectrum".*

Done.

*Line 166. Omit "the" at end.*

Done.

*Line 203. You could omit ", which rotated in the direction of the Earth's rotation,"*

Done.

*Line 204. If "within ~1 day" refers to "dissipated quickly" rather than how quickly the wind forcing stopped then better "dissipated quickly, within ~1 day after the wind forcing stopped. This short duration . . ."*

Thanks, and done accordingly.

*Line 208. I do not understand "back". Omit?*

Done.

*Line 229. ". . out of the"*

Done.

*Line 285. "smaller increasing rate" -> "smaller rate of increase"*

Done.

*Line 305. "constrain" -> "constraining"*

Done.

*Lines 452-453. ". . . meantime upwelling, caused . ." [meantime one word; move ","]*

Done.

*Figure 8 caption. Line 2: "read" -> "red". Lines 3-4. I am puzzled; the orange curve arrows look like positive relative vorticity, the blue curve arrows look like negative relative vorticity, the opposite of what you state for "zeta". Line 4: "yellows" -> "arrows".*

We replot the arrows.

*Figure 12 caption. Lines 2-4 refer to panels (d-f), (g-i), (j-l) but there are only panels (a-c) in the figure.*

The caption of Figure 12 is revised.

Thanks a lot for spending time to provide the very careful editing and comment.